# Mobility Prediction-Based Optimisation and Encryption of Passenger Traffic-Flows Using Machine Learning

**DOI:** 10.3390/s20092629

**Published:** 2020-05-05

**Authors:** Syed Muhammad Asad, Jawad Ahmad, Sajjad Hussain, Ahmed Zoha, Qammer Hussain Abbasi, Muhammad Ali Imran

**Affiliations:** 1James Watt School of Engineering, University of Glasgow, Glasgow G12 8QQ, UK; Sajjad.Hussain@glasgow.ac.uk (S.H.); ahmed.zoha@glasgow.ac.uk (A.Z.); Qammer.Abbasi@glasgow.ac.uk (Q.H.A.); Muhammad.Imran@glasgow.ac.uk (M.A.I.); 2School of Computing, Edinburgh Napier University, Edinburgh EH10 5DT, UK; J.Ahmad@napier.ac.uk

**Keywords:** transportation, RFID sensors, artificial intelligence, mobility predictions, optimisation, encryption, smart city planning, passenger pathways, machine learning, 5G

## Abstract

Information and Communication Technology (ICT) enabled optimisation of train’s passenger traffic flows is a key consideration of transportation under Smart City planning (SCP). Traditional mobility prediction based optimisation and encryption approaches are reactive in nature; however, Artificial Intelligence (AI) driven proactive solutions are required for near real-time optimisation. Leveraging the historical passenger data recorded via Radio Frequency Identification (RFID) sensors installed at the train stations, mobility prediction models can be developed to support and improve the railway operational performance vis-a-vis 5G and beyond. In this paper we have analysed the passenger traffic flows based on an Access, Egress and Interchange (AEI) framework to support train infrastructure against congestion, accidents, overloading carriages and maintenance. This paper predominantly focuses on developing passenger flow predictions using Machine Learning (ML) along with a novel encryption model that is capable of handling the heavy passenger traffic flow in real-time. We have compared and reported the performance of various ML driven flow prediction models using real-world passenger flow data obtained from London Underground and Overground (LUO). Extensive spatio-temporal simulations leveraging realistic mobility prediction models show that an AEI framework can achieve 91.17% prediction accuracy along with secure and light-weight encryption capabilities. Security parameters such as correlation coefficient (<0.01), entropy (>7.70), number of pixel change rate (>99%), unified average change intensity (>33), contrast (>10), homogeneity (<0.3) and energy (<0.01) prove the efficacy of the proposed encryption scheme.

## 1. Introduction

The current exponential passenger traffic flow is a precursor towards an imminent traffic flux, encryption, and capacity crunch. In this backdrop, effective management of traffic through optimisation, preserving confidential data streams and effective utilisation of resources through deployment of a large number of 5G Heterogeneous cells (HetNets) in the train underground environment have emerged as the most conceding solution to achieve the prediction accuracies, encryption, and manifold capacity gain goal [1]. However, traffic flows and encryption of the passengers’ data are on a direct collision path with complicated passengers traffic movement, which needs AI for optimisation, encryption, and an energy-efficient vision of 5G HetNets deployment in the train underground environment. This is due to the complexity in high accumulated traffic flows with relevant information, i.e., traffic patterns within the LUO environment, to be optimised [1,2], and encrypted [3]. It is difficult to understand the LUO ecology from a business point of view when cellular services are limited [4]. Furthermore, traffic variations are unknown, which is one of the dominant factors that affect the environment. With the limited information, optimisation and encryption would become significantly challenging, which eventually leads to ineffective resource management and a high number of unnecessary deployments that have ${\mathrm{CO}}_{2}$ emissions [3,5] and costs [6].

### 1.1. Motivations

With the promise of 5G cellular networks, the aim is to cover virtually every market that comes under SCP. The broader reach will transform everything from existing technologies to automotive functionalities of mobile communication, train signalling, logistics, automated complex encryptions, Train to Train (T2T) communication, and large-scale businesses. There are many other applications of 5G for immersive training and experiences useful for complex tasks vs. traditional counterparts. Motivations behind this work are outlined as:Real-Time processing (Train Network side): Engineers and control centres would be able to remotely access the network for maintenance purposes such as passenger traffic flows for undertaking any measures against safety, monitoring passengers’ pathways to advise the best possible routes in real-time and reduce the risk of critical conditions. In the context of the LUO environment, there is an immediate requirement to optimise the whole network with train conditions and live monitoring for improving attribution of delays, scheduling, and analysis. Furthermore, it would enhance train prediction times, which is the cornerstone of better passenger journeys, suggesting alternate routes and positioning on the correct platform at terminus stations.Passenger live experience (User side): 5G can help facilitate daily passengers to avoid such trains which are congested and have minimum space to comfortably board on. 5G automated mechanisms, such as sensors within the train carriages, tunnels, stations and, platforms, would assist passengers in deciding the best pathway to take.Low latency for real-time data response: Manage customer incidents to reduce the risk of antisocial behaviour and to improve passenger safety, a 5G empowered mechanism using the AEI framework would be able to provide better automated incident management.Data and Analytics: For increasing the revenue through media and advertising, it would attract companies to advertise on the train network when demographics and AEI traffic flow information through predicted cases are available. Similarly, train braking performances are recorded through moisture sensors, which is an essential part of adhesion management and control. Hence, using the AEI framework with the mobility prediction accuracies, braking rate adjustments and reduction in unnecessary delay information can be obtained. This would, in turn, automate the braking adjustments while continuously monitoring predicted real-time traffic flows through proactive scheduling decisions. Furthermore, weight and temperature monitoring are other applications equally important when considering the train network below ground. Therefore, 5G empowered mechanisms that would monitor predicted weight and temperature in the real-time scenario, advising passengers to move into other train carriages, removing passengers from specific carriages, providing real-time advice to carry water in hot temperatures, de-training in critical circumstances, enforcing no-trains if delays are possible, holding doors for longer periods, providing better information to reduce risk of carrying ill passengers, minimizing platform crowding, and providing accurate timetabling and scheduling, would further strengthen the theme of mobility predictions.Private AEI Framework: To preserve real-time passenger data recorded through the tap-in tap-out machines at the stations, real-time encryption is required to provide an added security layer. Lightweight encryption with less time of operation is important against malicious attacks on the key information that is subject to authorised staff only.

The proposed solution is designed to monitor traffic movement using the AEI framework that exploits mobility prediction classifications and data encryption requirements accordingly. The focus of this work is to analyse AEI data and provide possible solutions on the aforementioned points to optimise the network.

### 1.2. Related Work

Mobile operators capitalises on large scale mobility traces obtained via mobile phones to optimize their network operational behaviour. In addition, movement patterns aid to develop a scenario where mobile users play a significant role in their movement behaviours through most visited places. Embracing the world in the 5G era, challenges in understanding traffic patterns and human mobility predictions is limited, which has been experienced by urbanised cellular towers [4,7,8]. Extra diligent intelligence is required to fully automate the network that would be a hypothetical remedy against current complexities involved in predicting the traffic flows in the underground train environment [9,10]. Traffic flow patterns can be predicted by the ML models by using historical data, as shown in Figure 1. This would aid in developing solutions for optimisation and encryption, which makes it easier to proactively monitor and predict passengers movements [9,11].

Several studies on real-time datasets despite randomness in the path of traffic flow, show user movement predictions [12,13]. The movements are user trajectories from source to destination with regular intervals, expected or unexpected. Several comprehensive surveys for mobility prediction are available in [14,15,16] that exploit various methods of predicting user mobility patterns where Markov chain-based is a popular predictor due to being less complex in nature [12,15,17]. However, with the limitations of real-time datasets that have complexity involved, ML predictors can be a viable alternative in order to study traffic flow and provide encryption to it. Hence, there is a need to explore the performance of ML predictors for the mobility prediction by using a meaningful and novel AEI framework that, with the best of our knowledge, has never been considered before. Several mobility prediction schemes probabilistic approaches for predicting the likelihood of the next destination were discussed in [16,18,19,20]. In the context of cellular network optimisation, various machine learning algorithms including decision tree [21], k-means algorithm [22,23], and artificial neural networks [24] have been employed for predicting user mobility patterns in order to perform network resource optimisation patterns. In the context of road networks, NN algorithms [25,26] have been used for short-term traffic flow prediction, as well for reduce road congestion by analysing traffic information and further relaying the message back to the vehicles [27]. A smartphone based software to recognise traffic flows with high accuracy was proposed by using the Random Forests (RF) classification model and positioning technology in [26,28].

London Underground Limited (LUL) is one of the oldest and largest tubes in the world, which is renowned for carrying a large number of passengers in all cardinal directions of the metropolitan city, London. Due to the nature of the trains, they are heavily loaded, and it is of utmost importance that user-friendly traffic information within train carriages to prevent superfluous congestion and confusion is provided. It is evident from the fact of the horde of passengers travelling at different times of the day must be facilitated with comfort and safety [2,29,30]. A similar contribution was made in the field of train adaptive control when identifying the rolling stock parameters [29] of moving trains. Furthermore, an article was proposed that focused on the paradigm of localisation of rolling stocks and its movements, dispatching control, and rail traffic in [30]. A system-theoretic standpoint for establishing transformation and reduction of the parallel paths; reducing overhead was developed in [31]. A similar study was contingent upon robustness into the operational system by analysing the problems of rail transformation of the network to some parallel lists where a taxonomy of the time-optimality criterion is proposed for an ordinary differential [32]; however, the transformation of the railway does not take passenger traffic flows into account through which robustness can be injected into the operation of the rail system. Another illustration in regard to correlations of rail transit traffic flow, which impacts the train control system on rail transit service quality [33].

Some works are found in the context of passenger traffic flows, network complexity, and energy efficiency [34]. Real-time traffic information that drives interstation running time monitored by train supervision systems in [35]. In the same context, another work that further classifies the traffic into weights and temperatures is found where the structure of the classification training model was proposed with an ML algorithm, such as KNN in [36]. The cellular coverage inside the underground stations are often patchy or in some cases non-existent whereas the traffic flows are quite complex. This makes it quite challenging to control and predict the passenger flow variations and also real-time operational optimisation for smart city planning (Mayor of London Transport Strategy can be found online at: https://www.london.gov.uk/sites/default/files/mayorstransport-strategy-2018.pdf).

Every day, thousands of passengers use RFID cards to tap in and tap out of the train stations providing an estimate of passenger flows. The use of passive RFID technology requires an RFID card and a reader that is cost-effective and secure based on Radio Frequency (RF) electromagnetic fields. RFID operates in the range of frequencies such as; Low-Frequency (LF) runs at 125 to 134 kHz, High-Frequency (HF) at 13.56 MHz, and Ultra-High-Frequency (UHF) runs at 433 and 860–960 MHz. The database linked to RFID devices stores the passenger data in the form of unique identification numbers through an electronic microchip. In the context of encryption, some works have been conducted to alter the data in such a way that it appears random and irregular [37]. Two types of encryption, known as symmetric key and asymmetric key in [37] are used to highlight the importance of current trends in encryption. In the symmetric key algorithm, keys at encryption and decryption are the same level, while they are different in asymmetric keys. In both types of algorithms, the main aim of encryption is to protect the valuable data from attackers. Chaotic systems can produce random data that can be employed in a cryptosystem [38]. In [38,39,40,41], researches have reported a number of encryption schemes that used chaotic maps, which are well suited for light weight encryption and offer ergodicity, sensitivity, and randomness. [38,39,40,41]. In our work, we have also used the aforementioned properties of chaos and proposed a scheme for protection data from attackers. Two chaotic maps, nonlinear chaos map [42] and Logistic map [43], are used in the encryption process, which is discussed in more detail in the latter part of the paper.

In this direction of research, existing works, but not limited to, show reasonable results in relation to above-ground traffic movements, optimisation of traffic in an urban city environment, and encryption of simple data. However, to the best of our knowledge, existing contributions discussed in the context of mobility predictions and encrypting valuable real-time data approaches fall short of the mark for 5G requirements due to following six limitations:Reactive mode of operation: Traditional SON algorithms are reactive in nature and the methods employed for mobile network optimisation are not well suited in the context of the target problem since passenger traffic flows in an LUO environment are dynamic and constantly varying. Improvement can be obtained through this method but at the cost of sacrificing time, resources, and QoS. However, due to the continuously varying dynamics of the passenger traffic flows in an LUO environment according to time of congestion on the platforms and stations when a remedy is planned, the conditions may have already changed drastically. This leaves a gap in planning new remedies before it can be influenced. The problem becomes worse in 5G, where complexity of haphazard assortment of different types of passengers traffic in either absence or limitations of cellular coverage within the LUO environment.5G optimisation in ultra reliable low latency: Real-time alerts, monitoring, and supporting mission critical applications are required to meet certain 5G optimisation and latency standards [44] keeping good QoS and without affecting the operational technology (OT) train network. Traffic complexity on stations, tunnels, and platforms add unnecessary latency, which puts the train’s operational network in a difficult position to address mission critical applications. Therefore, a demand for predicting passenger flows for low-latency remedies demands is needed.User Flow Discovery in LUO Environment: A key challenge to discover a user pattern where users have multiple ways to travel in the LUO network, such as Access, Egress, and Interchange (AEI), along with the ridership data obtained from Interchange-Alighters and Interchange-Boarders. Existing mobility prediction methods overlook this challenge to the best of our knowledge. User mobility pattern approaches may work in low, medium, and high density networks above ground where the LTE cellular network is available; however, we are not aware of any studies that address the problem of 5G scalability, measurability, and applications in complex LUO ecology.Intelligent transport systems (ITS): Another challenge in the 5G domain is to have an intelligent system that would assist transportation in SCP. Many concepts have been proposed to regulate the mobility of users above ground by using cellular services. However, there is not much work done in the field of ITS using the AEI framework in an LUO environment where cellular services are patchy. With the limitation of cellular services, either on-board train modules or ticket machines take the responsibility of traffic flow monitoring. The 5G concept of onboard ITS is fairly new, which is yet to be deployed. Train suppliers, for example, Siemens, are making splendid efforts in order to deliver innovative trains with special functionality of on-board monitoring concept (Mobility in Metro London can be found online at: https://www.mobility.siemens.com/global/en/portfolio/references/metro-london.html).Planning and cost of technology: When 5G brings numerous benefits to the technology, it also brings concerns over planning and deployment costs. There are various methods discussed within the domain of 5G, associated with planning and costs in the energy efficiency, densely populated HetNets, spectrum usage domain, internal logistics and Logistics 4.0, transport systems, etc. However, there seems to have been less work conducted in the domain of classification of mobility predictions and encryption modelling considering passenger traffic flows in underground trains.Encryption: Advanced Encryption Standard (AES) and Data Encryption Standard (DES) can provide confidentiality but for real-time encryption, a light-weight encryption algorithm is required [42]. Over several years, cryptographers are using chaos-based cryptosystems for faster and real-time encryption. In this paper, we have also used two chaotic maps known as a nonlinear chaotic map and logistic map that have quick time responses and have lower memory sizes compared to existing schemes. Our novel scheme would be able to provide an extra layer of security that is difficult to deduce secret cryptographic keys. One can also propose an encryption algorithm with a single map for faster processing but due to lower key space issues, we have used two maps in this research.

### 1.3. Contributions

To address the aforementioned limitations, we propose a novel AEI based optimisation and encryption framework, as shown in Figure 1. The aim is to make emerging cellular and train systems artificially intelligent and autonomous in order to anticipate and encrypt user mobility behaviour within the LUO environment. The intelligence obtained from the aforementioned framework can help streamline near-real time operational optimisation. This includes minimization of congestion at the interchanges, optimal resource scheduling while proactive encryption schemes make sure that passenger data privacy is preserved. The contributions and organisation of the paper can be summarized as follows:As a building block of the AEI framework, we propose ML driven models that take into account spatio-temporal characteristics of passenger flows in the LUO environment for mobility prediction in a large-scale train network. Our proposed mobility prediction model overcomes the limitation of conventional ML classification algorithms that failed to incorporate high accumulated passenger traffic in three dimensional states (3D), i.e., number of passengers, travelling time, and AEI based passengers travelling and behavioural information (Section 2.1 and Section 2.2).Based on the intelligence gained from the mobility model, i.e., mobility prediction classification and directions, a proactive movement precision is formulated to maximise the advantage of traffic flows in several unexpected directions and instructing passengers to take necessary interchanges. In this way, real-time directions can be exploited for monitoring purposes shown in Section 2.3. Classification estimation for the next passenger movements is mentioned in Section 2.4.We also propose a novel encryption method to preserve real-time passenger traffic flows where a system incorporates cost, easy deployment, security and privacy preservation aspects (Section 2.5). This is benchmarked against the current security parameters and measures that have been used across transportation specifically in train ticket machines using RFID technology. The encryption provides security, which is transaction oriented data integrity that is light weight, proactive, and provides faster data rates than existing technologies.Next, we propose a novel method to map the classification results through comparative performance analysis of six ML algorithms, comprehensively. It has been shown that the highest prediction accuracy has been obtained by ANN, as detailed in Section 3.3.1. In addition, an encryption algorithm that is capable of handling the heavy passengers traffic flow in real-time while providing faster processing that can hold an unlimited number of different applications without any limitations of memory sizes is discussed in Section 3.3.2.System level comprehensive performance analysis of our proposed model have been conducted that complies with multi-tier 3GPP simulations. The prediction accuracies of ML algorithms have been compared using a realistic AEI framework. Error margins have been estimated in cross validation of training real-time data to be around 10%.

## 2. System Model

An approach to optimise the train network, we present an analytical model development of AEI framework in which the foundation is based on the following elements:Artificial Neural Network (ANN) driven Mobility Prediction.Movement Precision to map Future User Location.Next Movement Classification Estimation.Encryption based real-time security built into the passenger traffic flow recorded by RFID contactless devices at ticket machines.

### 2.1. AEI Framework

The AEI framework proposed in this paper only focuses on the real-time data of train network systems that covers all LUO stations’ passenger flows. Consideration has been given to the total number of passengers movement in a given period of time that have been classified into access, egress, and interchange models. All active train lines moving in North, West, East and South directions are fully operational without any problems. Directions of all passengers using the LUO network are known on the principle of their positioning recorded by tap-in and tap-out technology. Assumptions have also been taken into consideration where all passengers use the provided technology without any malfunction. In other words, a model based on full buffer mobility traffic flow is used for each traveller, i.e., there is no defect in data availability and it is always present to be monitored with a constant bit rate. For proactive optimisation and encryption, a centralised smart transportation architecture is assumed. Furthermore, intersection traces that encompass past and future platform’s time and location stamped information, such as start and end platform nodes for a particular train line, in-use station’s logistic code for all train lines, and all other nodes transitions not in use, are known and assumed to be available to the smart transportation server. Alongside, a method of encryption is used in the framework that addresses the issue of data privacy while ensuring faster data transfer. Through this, we would proactively preserve AEI network state information using automated deep learning prediction models with the help of encrypted images.

### 2.2. Artificial Neural Network (Ann) Driven Mapping of Mobility Prediction

ANN is an interconnected group of nodes/neurons consisting of input and output layers. Based on the training data, these neurons learn the input–output mapping without being programmed with task-specific rules. In other words, numeric weights are designed in such a way that they can be tuned based on experiences to exploit the best possible outcomes when the neural nets show flexibility to inputs and intelligence when learning. We have implemented an ANN model to classify passenger traffic flow patterns in the LUO environment and used that to predict future location based on our complex AEI dataset. The movement patterns of 3D layers (number of passengers, time of travelling and AEI) and the relationship between inputs and outputs is described in the following modified equations:(1)$$p\left(y_{t}=c|w\subset D\right)=w_{o}+\sum_{j=1}^{Q}w_{j}\cdot f\left(w\right),$$
(2)$$f\left(w\right)=f\left(w_{0,j}+\sum_{i=1}^{P}w_{i,j}\cdot \;x_{t-i}\right),$$
where; *p* is probability of three classes *c* within our AEI dataset *D*, which depends on $x_{t-i}$ as inputs (1, 2, …, *P*), $w_{i,j}$ (i=0,1,2,…,P; j=1,2,…,Q), $w_{j}$ is (0,1,2,…,Q) are connection weights. *P* and *Q* are represented as input and hidden nodes, $y_{t}$ is the output that depends on integer *t* transition of layers from 0 to 2 of the indices in dataset *D*, and f(w) is the transfer function that depends on the number of weighted nodes. There are various functions used in ANN such as; linear, logistic, quadratic, hyperbolic, and gauss. The most common function used in hidden layers is called logistic function. Therefore, for producing the best possible outputs, ANN performs a relationship among its inputs and outputs through nonlinear functionality, which is shown in the equation below:(3)$$y_{t}=F\left(x_{t-1},\ldots ,x_{t-P},w\right),$$
where, *w* represents connection weights as a vector and function *F* is nonlinear based on the parameters and structure of the network. In our study, ANN outperforms against all the other algorithms discussed for mobility predictions. Therefore, we have shown detailed ANN mapping as the best classification algorithm in the following section. We have established that the passenger positioning below ground and the number of passengers in a given period of time are the most important parameters in the mobility prediction schemes. We envisage that in the future, with the deployment of 5G HetNets in the LUO environment offering better cellular coverage and capacity, the accuracy of the ANN driven prediction models will improve if they take into account mobility traces as an additional input.

The ANN-based mobility model is trained by a large number of inputs called training samples associated with different traffic flow categories. In addition to the training matrix, the weights and biases are adjusted to satisfy the ANN mechanism for proper mapping of inputs (AEI) and outputs (prediction accuracy) and adapt to new passenger positioning according to three classified classes. The AEI-based topological mapping of ANN is shown in Figure 2 to minimise errors that occurred by locations associated with the optimisation of weights and biases.

We define *w*, *x*, and *y* as weights, inputs, and outputs. Layers *l* are denoted as 0, 1, 2 for input, hidden and output layers, $w_{i,j}^{l-1,l}$ represents the relationship of weight’s connection through input, hidden and output layers, $x_{i,d}^{l}$ represent the input values from our dataset *d* with the perceptron *i* and *l* = (0, 1) is the transition layer from input to hidden layer. The parameter $y_{j,d}^{l}$ represents the output values with the perceptron *j* and *l* = (1, 2) is the layer transition from hidden to the output layer for our sample dataset *d*. Following are the equations that satisfy the relationship for each perceptron to classify number of inputs, their weighted transitions, and best predicted outputs.
(4)$$\begin{array}{cc} x_{i,d}^{l} & =f\left(y_{j,d}^{l}\right)=\lim_{y(0\to 1)}\frac{1}{1+exp(-y_{j,d}^{l})},\\ y_{j,d}^{l} & =\int_{j}\sum_{i=1}^{D(l-1)}\left(w_{i,j}^{l-1,l}\;x_{i,d}^{l-1}\right)-\theta_{j}^{l-1,l}\;dx,\\ D(l);\;j & =\left\{\begin{array}{cc} D, & l=0,\\ T, & l=1;\\ j, & l=1,when\;j=1,\ldots T,\\ j, & l=2,when\;j=1\;or\;2, \end{array}\right. \end{array}$$

The optimal weight and bias calculation is performed by iterating several times to reach to a optimum state where training error is the minimum. The error minimisation equation is as follows.
(5)$$\Delta_{\left(y,z\right)}=\frac{1}{2}\left[\sum_{d=1}^{D}\sum_{j=1}^{2}{\left(y_{j,d}^{2}-z_{j,d}\right)}^{2}\right]$$
where, $\Delta_{\left(y,z\right)}$ is the difference margin of training error depends on $y_{j,d}$ and $z_{j,d}$ vectors. *D* is our dataset training samples, $y_{j,d}$ represents the output of *x* and *y* coordinates and, $z_{j,d}$ denotes the training phase coordinates with the expected *x* and *y*. Utilisation of the back-propagation (BP) method would essentially provide the essence of neural net training in order to obtain optimal weights and biases by practicing fine-tuning. Following are the equations from [45] for optimal weights and biases tuning, which are modified according to our AEI framework.
(6)$$\begin{array}{cc} w_{i,j}^{l-1,l}\left(k+1\right) & =w_{i,j}^{l-1,l}\left(k\right)-\frac{\alpha \partial \Delta_{\left(y,z\right)}}{\partial w_{i,j}^{l-1,l}\left(k\right)},\\ \; & =w_{i,j}^{l-1,l}\left(k\right)-\alpha \int_{j}\sum_{d=1}^{D}\delta_{j,d}^{l}\left(k\right)\;y_{j,d}^{l-1}\left(k\right)\;dy,\\ \theta_{j}^{l-1,l}\left(k+1\right) & =\theta_{j}^{l-1,l}\left(k\right)-\beta \int_{j}\sum_{d=1}^{D}\delta_{j,d}^{l}\left(k\right)\;y_{j,d}^{l-1}\left(k\right)\;dy, \end{array}$$
(7)$$\begin{array}{c} \delta_{j,d}^{l}\left(k\right)=\left\{\begin{array}{cc} \overset{´}{f}\left[x_{i,d}^{l}\left(k\right)\right]\sum_{m=1}^{2}\delta_{m,p}^{l+1}\left(k\right)\;w_{j,m}^{l,l+1}\left(k\right), & l=1,\\ {}[y_{j,d}^{l}\left(k\right)-z_{j,d}]\overset{´}{f}\left[x_{i,d}^{l}\left(k\right)\right], & l=2, \end{array}\right. \end{array}$$
where, *i* is (1,…,N) when l=1 and (1,…,T) when l=2. Similarly, *j* is (1,…,N) when l=1 and (1 or 2) when l=2. Number of maximum iterations are represented by k≤K. We define α and β as rate of learning weights $w_{i,j}^{l-1,l}$ and biases $\theta_{j}^{l-1,l}$.

### 2.3. Movement Precision to Map Future User Location

When aiming to establish optimal movement patterns while their location precision in LUO environment, it is undoubtedly a challenge to come up with an accurate outcome that would comply with all geometrical parameters. Therefore, we have used an ANN-based precision matrix for underground stations, platforms, and tunnels to analyse relative distances and recorded AEI information received from different ticket barriers using tap-in and tap-out technology. TfL train lines called, Jubilee Line and London Overground are chosen for the test scenario on one of the stations in London that has several access, egress, and interchange points, as shown in Figure 3.

We calculate the mean $\bar{M}$ values from different ticket barriers through which passengers get in and out of the stations to terminate or carry on their journeys. Now, the mean for AEI framework would be calculated with the help of passengers access *A*, passenger egress *E* and, passenger interchanges *I*, being the three correlation factors to determine the precision coordinates.
(8)$$\begin{array}{cc} A_{k}\left(x,y\right) & =A_{k,x,j}+A_{k,i,y}\;,\\ A_{k}\left(x,y\right) & =\frac{\int_{j}\sum_{i=1}^{n}\left(M_{i,j}-\bar{M_{i,j}}\right)\left(i-\bar{i}\right)\;dj}{\sqrt{\sum_{i=1}^{n}{\left(M_{i,j}-\bar{M_{i,j}}\right)}^{2}\sum_{i=1}^{n}{\left(i-\bar{i}\right)}^{2}}}+\;\frac{\int_{i}\sum_{j=1}^{n}\left(M_{i,j}-\bar{M_{i,j}}\right)\left(j-\bar{j}\right)\;di}{\sqrt{\sum_{j=1}^{n}{\left(M_{i,j}-\bar{M_{i,j}}\right)}^{2}\sum_{j=1}^{n}{\left(j-\bar{j}\right)}^{2}}}, \end{array}$$
(9)$$\begin{array}{cc} E_{k}\left(x,y\right) & =E_{k,x,j}+E_{k,i,y}\;,\\ E_{k}\left(x,y\right) & =\frac{\int_{j}\sum_{i=1}^{n}\left(M_{i,j}-\bar{M_{i,j}}\right)\left(i-\bar{i}\right)\;dj}{\sqrt{\sum_{i=1}^{n}{\left(M_{i,j}-\bar{M_{i,j}}\right)}^{2}\sum_{i=1}^{n}{\left(i-\bar{i}\right)}^{2}}}+\;\frac{\int_{i}\sum_{j=1}^{n}\left(M_{i,j}-\bar{M_{i,j}}\right)\left(j-\bar{j}\right)\;di}{\sqrt{\sum_{j=1}^{n}{\left(M_{i,j}-\bar{M_{i,j}}\right)}^{2}\sum_{j=1}^{n}{\left(j-\bar{j}\right)}^{2}}}, \end{array}$$

Since the interchange precision is a combination of multiple points’ coordinates, which include all the interchange-alighters, interchange-boarders, and all the possible movements, $f(IP_{i,j}^{k})$, we have following equations which would provide accuracy in capturing all the possible movements under interchange. We aim to provide more complexity in the mathematical model by using point-to-point and point-to-plane distances considering velocity and displacement in future work.
(10)$$\begin{array}{cc} I_{k}\left(x,y\right) & =I_{k,x,j}+I_{k,i,y}+IP_{i,j}^{k}\;,\\ I_{k}\left(x,y\right) & =\frac{\int_{j}\sum_{i=1}^{n}\left(M_{i,j}-\bar{M_{i,j}}\right)\left(i-\bar{i}\right)\;dj}{\sqrt{\sum_{i=1}^{n}{\left(M_{i,j}-\bar{M_{i,j}}\right)}^{2}\sum_{i=1}^{n}{\left(i-\bar{i}\right)}^{2}}}+\;\frac{\int_{i}\sum_{j=1}^{n}\left(M_{i,j}-\bar{M_{i,j}}\right)\left(j-\bar{j}\right)\;di}{\sqrt{\sum_{j=1}^{n}{\left(M_{i,j}-\bar{M_{i,j}}\right)}^{2}\sum_{j=1}^{n}{\left(j-\bar{j}\right)}^{2}}}+\\ \; & \frac{\int_{i,j}\sum_{i,j=1}^{n}\left(M_{i,j}-\bar{M_{i,j}}\right)\left(i-\bar{i}\right)\left(j-\bar{j}\right)\;d\left(i,j\right)}{\sqrt{\sum_{i,j=1}^{n}{\left(M_{i,j}-\bar{M_{i,j}}\right)}^{2}\sum_{i,j=1}^{n}{\left[\left(i-\bar{i}\right)\left(j-\bar{j}\right)\right]}^{2}}}, \end{array}$$
where, $M_{i,j}$ is the mean value of the traffic flow provided by LUO from multiple reference points for kth iterations and $\bar{M_{i,j}}$ represents $M_{i,j}$ mean values. $A_{k}\left(x,y\right)$, $E_{k}\left(x,y\right)$ and, $I_{k}\left(x,y\right)$ denotes access, egress and interchange coordinates on *x*-axis and *y*-axis. We define, $\Gamma_{k}\left(x,y\right)$ which is the function to provide relative *x*, *y* values according to AEI framework from Equations (Equation 8)–(Equation 10).
(11)$$\Gamma_{k}\left(x,y\right)=A_{k}\left(x,y\right)+E_{k}\left(x,y\right)+I_{k}\left(x,y\right),$$

### 2.4. Next Movement Classification Estimation

For the purpose of classification estimation using our novel AEI framework, we use softmax function which is ANN network-based classifier in the final layer. The aim is to train the AEI information under cross-entropy approach [45] for optimal results. In mapping two probability distributions *p* and *q*, the set of events can be identified that measures average number of bits vital for identification of drawn events from the given set of underlying set of events. Here, *p* is true distribution and *q* is an estimated probability distribution. We define the distribution using both probability distributions *p* and *q* for our dataset as:(12)$$\begin{array}{cc} \; & f\left(d\right)\;\left[\eta_{k}\left(p,q\right)\right]=-\sum_{x\in X}\;p\left(x\right)\;log\;q\left(x\right),\\ \; & f\left(c\right)\;\left[\eta_{k}\left(p,q\right)\right]=-\int_{X}\;P\left(x\right)\;log\;Q\left(x\right)\;dr\left(x\right), \end{array}$$
where, $\eta_{k}\left(p,q\right)$ is the entropy function, *X* being the dataset in a precise notion in both discrete and continuous distributions. In particular, for continuous distribution, an assumption has been made that *p* and *q* are absolutely continuous associated with their reference measures *r*. Using Equation (Equation 12), we now contribute to use $\eta_{k}\left(p,q\right)$ for three different variables as:(13)$$\begin{array}{cc} \; & f\left(d\right)\;\left[\eta_{k}\left(p,q\right)\right]=-\sum_{x\in X}\sum_{y\in Y}\sum_{t\in T}\;P_{k}\left(x,y,t\right)\;log\;Q_{k}\left(x,y,t\right),\\ \; & f\left(c\right)\;\left[\eta_{k}\left(p,q\right)\right]=-\int_{X}P_{k}\left(x,y,t\right)\;log\;Q_{k}\left(x,y,t\right)\;dr\left(x,y,t\right), \end{array}$$
where, *x*, *y*, and *t* are number of passengers, AEI dynamic information, and travelling time.

### 2.5. Encryption of Passenger Traffic Flows

Access, egress, and interchange is highly important information, and an eavesdropper(s) can access this sensitive information for any means. Figure 4 shows plain text data that can reveal important information regarding the AEI framework, which, through the following method, would be encrypted to form a designated encrypted model. We define a novel encrypted model using the below algorithm. Detail steps are outlined as:Let *A* is a plain text data having size A×B. Apply secure hash algorithm (SHA-512) on *A* and get a 128 hexadecimal value. Store SHA value in ψ.Convert ψ into decimal and store value in ω.Get an initial value for chaos map using the below equation:
(14)$$x_{n}=\frac{\omega }{2^{512}}$$Provide $x_{n}$ seed parameter to Nonlinear chaos map given below [42]:
(15)$$x_{n+1}=\left(1-\beta^{-4}\right)\cdot \cot\left(\frac{\alpha }{1+\beta }\right)\cdot {\left(1+\frac{1}{\beta }\right)}^{\beta }\cdot \tan\left(\alpha x_{n}\right)\cdot {\left(1-x_{n}\right)}^{\beta },$$
where the seed parameters are defined as:
$$\left\{\begin{array}{c} x_{n}\in \left(0,1\right)\\ \alpha \in (0,1.4]\\ \beta \in [5,43] \end{array}\mathrm{or}\begin{array}{c} x_{n}\in \left(0,1\right)\\ \alpha \in (1.4,1.5]\\ \beta \in [9,38] \end{array}\mathrm{or}\begin{array}{c} x_{n}\in \left(0,1\right)\\ \alpha \in (1.5,1.57]\\ \beta \in [3,15] \end{array}\right.$$Define other seed α and β parameters for chaos map and iterate map 3×(A+B) to get random sequences and save sequence in Γ.Convert plain text information into three different channels, i.e., Ω, Ψ, and Φ. Now shuffle rows and columns of each channel with the sequence obtain from the chaos map and store value in $\Omega_{p}$, $\Psi_{p}$, and $\Phi_{p}$, respectively.Logistic map is written as [43]:
(16)$$y_{n+1}=r\left(y_{n}\right)\left(1-y_{n}\right),$$
In the above equation, $y_{n}\in$[0,1] and r∈[0,4] are initial conditions of map. Iterate Logistic map 3×A×B times and multiply the obtained value with $10^{14}$ and save the result in a row matrix *R*. Apply the modulus operator and save result in *S*:
S=Rmod(256).Reshape *S* into three separate matrices, i.e., $S_{1}$, $S_{2}$, $S_{3}$ and Apply XOR operation:
$$C_{1}=\Omega_{p}⊕S_{1}.$$
$$C_{2}=\Psi_{p}⊕S_{2}.$$
$$C_{3}=\Phi_{p}⊕S_{3}.$$Combine $C_{1}$, $C_{2}$, $C_{3}$ and save the value in *C* that is the final encrypted sensitive information.

## 3. Methodology

We present our results based on the novel AEI framework proposal where we first analyse the proactive ML-based automated classification of mobility prediction using an ANN-based algorithm. Second, we have used two chaotic maps known as nonlinear chaotic and logistic maps for encryption. An encryption algorithm with a single map is used for faster processing but due to lower key space issues, we have used two maps in this research work that can hold an unlimited number of different applications without any limitations of memory sizes due to their lightweight nature. The measured performance from the comparative analysis in the first part has been benchmarked against five algorithms (i) K-Nearest Neighbour (KNN), (ii) Support Vector Machine (SVM), (iii) Discriminant Analysis (DA), (iv) Naive Bayes (NB), and (v) Decision Tree (DT) by using 3D information including, number of passengers, time of travelling, and AEI. Three classes of AEI (access, egress, and interchange) have been used to classify the best possible mobility predictions. In the second part, we have benchmarked the best-chosen algorithm against classification modelling, movement precision, and classification estimation for future estimation.

### 3.1. Machine Learning Based Mobility Prediction Algorithms

ML was invented from pattern recognition and had the premise to automate intelligent machines that would essentially learn from and adapt to the associated environment through learned scenarios [24,46]. Due to the increasing measures of data beyond smart cities and communication networks and the necessity for intelligent data analytics, the use of ML algorithms has become a reasonable response to the challenging cases across many divisions such as; entertainment, social and financial services, entertainment, transportation, and health care. Using the discussed ML algorithms and features in order to organise movement patterns that reveal relationships and predict system dynamics or human behaviour, system operators can make automated intelligent decisions without any human intervention [47,48,49,50,51]. Passenger motion and their activities through ML bring advantages to the transportation sector where passenger movement is recorded through RFID technology. Some works on mobility predictions were developed for addressing energy saving and optimisation problems that proactively schedules resources, predicts future cell movements, and analyses the impact of cell load thresholds [5,18]. For the optimisation of passenger movements, ML algorithms proposed in [22,23,24,33] are used to describe traffic identification and classification of congestion patterns for problematic road segments. The study is based on traffic density and the average speed of vehicles where traffic parameters are recorded by sensors at various road segments. Automated ML mobility models and mechanisms have an immense impact on the performance of passenger traffic flows in general and, in particular, when the AEI framework is in the discussion. The passenger movement on the LUO network is observed and mobility models are used to determine their patterns and classify them in three dimensional (3D) states with respect to their headcounts, travelling time, and AEI travelling directions. Also, these models would manage to capture mobility in real-life applications. Although there are many ML mobility prediction models that can be discussed and compared in the research, we chose to focus on the best-performing algorithms according to our framework.

#### 3.1.1. K-Nearest Neighbour (KNN)

The first prediction mechanism we used is KNN, which is a non-parametric classifier. The function of KNN is such that it searches for K-points in its training set that are nearest to its test inputs, performs counting of its member classes, and returns observational fractions as estimated values [45]. The following modified equation is to be considered:(17)$$p\left(y=c|x\subset D,K\right)=\frac{1}{K}\sum_{i\in N_{K}\left(x,D\right)}I\left(y_{i}=c\right),$$
where, *p* is probability of classes $x_{i}$, which depends on $x_{1}$, $x_{2}$ and $x_{3}$ as test inputs within our AEI framework, $N_{K}\left(y,D\right)$ are the indices of the K nearest points *K* to an integer *y* in dataset *D* and I(e) is an indicator function when *e* is 0, for false and 1, for true. A fair example of memory-based learning is KNN algorithm, which is often called instance-based learning as well. Although other metrics can be used, the Euclidean distance metric is commonly used to limit the real-time data applicability. Here in our model, the input is three dimensional representing three distinct classes, and K=10. Simplicity of the KNN classifier depends on its labelled training data when it is provided with a good distance metric. KNN classifiers work fine with low inputs; however, they do not function nicely with inputs with high dimensions.

#### 3.1.2. Support Vector Machine (SVM)

The second mobility prediction mechanism we used is the SVM model, which is also recognised as a large margin classifier that classifies a set of inputs in the space of high dimensions through the liner and non-linear mapping. Therefore, predicted results are dependable only on a subset of the training data, known as support vectors and the modified loss function is known as a support vector machine or simply SVM [45]. The essence of the model revolves around decision boundaries to construct a hyperplane that produces distance bound nearest training samples. In our case, we used three classes of non-linear SVM with a radial basis function (RBF) kernel. Three samples $x_{1}$, $x_{2}$, and $x_{3}$, being the feature vectors in input space and kernel, representing AEI classes, are calculated as follows in the modified equations: (18)$$k\left(x_{1},x_{2},x_{3}\right)=exp\left(-\frac{{\left|\right|x_{1}-x_{2}-x_{3}\left|\right|}^{2}}{2\sigma^{2}}\right),$$
where, *k* is the kernel, σ is an RBF parameter. Since γ is $1/2\sigma^{2}$, then Equation (Equation 18) can be re-written as: (19)$$k\left(x_{1},x_{2},x_{3}\right)=exp\left(-\gamma {\left|\right|x_{1}-x_{2}-x_{3}\left|\right|}^{2}\right),$$

Based on the grid search, γ and *C* are optimised on training dataset subsets where γ is the new RBF and *C* being the SVM regularisation parameters. SVM performance is impacted from such parameters that have a high degree of association through subsets of training datasets, *k*, γ, and *C*. Type of algorithm’s kernel, SVM regularisation parameter *C*, and kernel coefficient parameter γ, is specified by the kernel, *k*. We used an SVM model where the RBF kernel is significantly important for determining the non-linearity function of hyperplane, *C* is set to default 1 value, and default value for γ as well. In our study, by setting up γ to a default value, the model outperforms at its best. Also, parameters like verbose, shrinkage, stopping criterion tolerance, probability, degree and max iteration were set to their default values. We used a kernel cache-size of 200.

#### 3.1.3. Discriminant Analysis (DA)

The third mobility mechanism we used is DA, which relies on independent variables to perform a set of prediction equations for the classification of individuals into groups. DA has two possible functional objectives where, (i) it classifies new individual inputs by finding predictive equations or, (ii) interprets the predictive equation to comprehend relationships that may be present among the variables. To a great extent, DA runs in parallel with multiple regression analysis (RA) but differs in variable discreteness, whereas RA deals with a continuous dependent variable. Due to the fact that our AEI framework’s dataset was dependent on discrete variables, we have only used DA for classification of discrete variables in our research. We have analysed mobility prediction accuracy of the equations by conducting residual investigation. Following is the modified equation from [45]:(20)$$p\left(y=c|x,\theta \right)=\frac{e^{\beta^{T}\cdot x+\alpha_{c}}}{\sum_{ci=0}^{T}e^{\beta_{ci}^{T}\cdot x+\alpha_{ci}}}$$
where, *p* is the exponentially distributed prediction equation of classes $c_{i}$ dependent on $x_{1}$, $x_{2}$ and $x_{3}$ as test inputs, *T* is an integer from 0 to 2, β an α are the indices in dataset θ.

#### 3.1.4. Naive Bayes (NB)

The fourth mobility mechanism we used is NB for classification of mobility predictions. This model shows, classification of vectors of discrete-valued features, $x\in {\left(1,...,K\right)}^{D}$, where *K* and *D* are the number of values for each feature and number of features [45]. We assume the classification features are conditionally independent with their class labels. In our case, we have $x_{1}$, $x_{2}$ and $x_{3}$ as class labels to indicate the AEI framework through which we have demonstrated class conditional density as a product, which is called the NB model. The modified one-dimensional density can be rewritten as:(21)$$p\left(y=c|x,\theta \right)=\prod_{i=0}^{I}\int p\left(y=c|x_{i},\theta_{ic}\right)\;dx,$$
where, *p* is the prediction equation of classes *c*, which depend on $x_{1}$, $x_{2}$, and $x_{3}$ as test inputs in dataset θ. Since the features are not independent in this model, it is termed as “naive”. In addition, class labels are not expected to be independent. Regardless, the assumption is not true, it still works well in classification due to the model’s simplicity.

#### 3.1.5. Decision Tree (DT)

The fifth mobility mechanism we used is DT, which is often called the classification and regression trees (CART) model. DT is a useful model that is defined by recursively partitioning the input space where each input space has a local model in each resulting region. As its name shows, the model can be represented by a tree with one leaf per region. Trees can be grown for the optimal partitioning of the dataset when required; hence, we use three classes as our inputs from the AEI framework in order to perform the classification of mobility predictions. The equation of the DT model is modified to meet our dataset requirement and is given below:(22)$$f\left(x\right)=\Xi \left[y|x\right]=\int \sum_{i=1}^{I}w_{i}\phi \left(x;v_{i}\right)\;dx,$$
where, Ξ[y|x] is the mobility prediction of three classes $x_{i}$ (*i* = 0, 1, 2) on the function f(x), $\phi (x;v_{i})$ is the *i*th region of the input classes, $w_{i}$ is the mean response, and $v_{i}$ provides coding of the variables choice, which split, and the threshold value, on the path from the root to the *i*th leaf [45]. This model defines regions and associated leaves due to the serving on the adaptability of a basis-function model. Weights in the model specify the response value in each region.

### 3.2. Simulation Settings and Data Set

We generated typical LUO-environment-based train traffic flow distributions leveraging 3GPP standard compliant algorithms that perform classification of network topology supported by simulations in MATLAB. The details of simulation parameters are given in Table 1, which explains our dataset for a week captured in the year 2017–2018, where *M* is referred to as a million counts.

In reality, modelling of traffic flow is based on real-time recorded data where passengers are distributed non-uniformly in the LUO areas such that the passengers were clustered around station, platforms, and tunnels in each of the three classes. We have used Monte Carlo style computational algorithms for simulation evaluations to establish the mean performance of the proposed framework. The selection of a mobility prediction based automated model was a real challenge when the objective was to represent the behaviour of passengers on the move in the LUO environment.

There are several models in the recent works in literature that provide mobility patterns of users end-to-end, and some inspiration can be taken from well known models such as Truncated Levy Walk, SMOOTH, SLAW [52]; however, in our case none of the mobility models best fit to the scenario. Based on deep analysis in relation to finding a close match, we came to a conclusion that some references can be used to define our AEI framework in light of SLAW (Self-similar Least Action Walk) [8]. References of the mobility model would be realistic when it exhibits real-time efficiency of passengers flow pattern, i.e., (i) truncated flights: the length of passengers flights, which are either straight line or haphazard trips with directional changes or pause, (ii) dissimilar mobility areas: passengers mostly move in their daily suited routes according to shortest times and less number of train line changes whereas different people may have different approaches when choosing routes, such as disability access, step-free access, etc., (iii) truncated inter-contact times: elapsed time between two stations including interchanges in-between by the same passenger, (iv) fractal way-points: passengers are used-to of their most visiting places and attractions, (v) convenient mobility areas: passengers are attracted to the specific travelling zones according to their daily budget. Therefore, the accuracy of the AEI framework is based on the mobility traces obtained from our dataset that already contained such information mentioned in proximity of SLAW. The mobility model was utilised to analyse parameters mentioned in Table 1. From the table, it can be seen that we have divided the data frame into multiple scenarios according to different times of the day, number of passengers in the given time, and classes associated to measure passengers in given time and traffic flow. The classification of mobility prediction interval was set to 21-h from 05:00 AM to 02:00 AM in our simulation as of negligible traffic flows monitored in the remaining hours of the 24-h day. The proposed encryption scheme is dependent on a nonlinear chaos map and Logistic map. The initial seed in the proposed work are $x_{n}$, α, β, *r*, and $y_{n}$. These seeds are all used as a key and must be kept secret from eavesdroppers. During simulation we have set $x_{n}$ = 0.1, α = 1.45, β = 10, *r* = 3.7, and $y_{n}$ = 0.001.

### 3.3. Results

#### 3.3.1. Mobility Prediction Accuracy

For ANN-based prediction accuracy benchmarking, our model is trained on 7 days of the week (the year 2017–2018) training data where we utilise Equations (Equation 4) and (Equation 6) to predict traffic flows dependent on weights and biases for every *l*, layer in *k* intervals. At each *k* interval, weights are observed and the accuracy of classification is then calculated by adding all the values in every layer for all time instants. Furthermore, our study is benchmarked against the movement precision of all the traffic in all the stations, platforms, and tunnels by using *x* and *y* coordinates for three classes; access, egress, and interchange. The calculations are reliant on the number of access points, the number of egress points and the number of interchanges within one station is represented as interchange points are obtained from Equation (Equation 11). Interchanges include passengers who alight on the same platform to take other train lines, passengers who alight on different platforms to take other train lines, and passengers who board onto the same train line but going in different directions from the same platform. All of these are dependent on the traffic movement at different times of the day (early, peak time travelers in AM and PM, passengers in midday, evening movements, and late-night traffic, as shown in Table 1.

We first present the simulation results obtained from our heavily loaded dataset, such as AEI dependent number of stations, number of passengers, and their time of travelling. This is shown in Figure 5. Now, we provide a classification analysis of mobility predictions by discussing six ML algorithms presented in this paper and present the performance of the best classification algorithm according to the prediction accuracies of high accumulated passengers traffic flows.

We have discussed ML algorithms and highlighted classifier libraries in Section 3.1. However, for details, we used a KNN classifier with *k* as the distance metric of nearest neighbours set to 3, SVM classifier with RBF kernel for the parameters setting where γ and *C* are set to default with the kernel size 200, DA with linear function, NB with normal function, DT with maximum splits set to 50. The rest of the values are set to default and to get optimum results, a single hidden layer of 10 neurons for ANN, and a ten-fold cross-validation was used for all the ML classifiers. A total of 677 observations were used for all three activities (Access, Egress, and Interchange) that were obtained from 05:00 AM to 02:00 AM (21-h). We have used accuracy as a performance metric for mobility prediction using the aforementioned classifiers, which are presented in Table 2. It can be observed that the NB algorithm provided the worst classification accuracy for all three classes providing an overall accuracy of 48%. The DT, SVM, and KNN algorithms performed almost similar, delivering overall classification accuracy of 80%. The ANN classifier performed better than all five classifiers by 10% where it provided overall classification accuracy of more than 90%.

The confusion matrices obtained using an ANN algorithm are presented in Figure 6, which demonstrates the overall classification of mobility prediction accuracies by using an ANN-based classifier in the AEI Framework for all the developed classes. The total number of available observations were divided into three parts, training, validation, and test datasets. To train the ANN algorithm, we have used 75% of samples (473) while the remaining 15% (102) for validation and 15% (102) for testing. Looking at Figure 6, we can see the number of correct observations for Class A (Access) are 69, which accounts for 75.8%. Similarly, for Class E (Egress) and Class I (Interchange), the correct classification is 90% and 87%, respectively. Moreover, as far as classification of validation is concerned, 63%, 88%, and 81% true classification rates are obtained for Class A, Class E, and Class I, respectively. In the dataset for testing the performance of the ANN classifier, 102 data samples were used. For class A, a total of 14 samples were used where 12 were correctly classified and 3 were misclassified. A total of 52 samples were used for Class E, where 49 were correctly identified and 3 were misclassified. Lastly, for Class I, 32 out of 35 samples correctly classified and 3 as misclassified. The overall test accuracy for all three classes while comparing all mobility prediction algorithms was obtained as 91.17% through ANN classifier, a best optimal result. Accuracy (%) is originally employed to evaluate the performance of all the algorithms in all possible scenarios using the AEI framework (i.e., for all train stations, passenger behaviours, and combination of their movements). The Receiver Operating Characteristic (ROC) curve is presented in Figure 7 for the best performing model (ANN) based on percentage accuracies in different passenger movement scenarios then metrics like precision (recall), Sensitivity (True Positive Rate (TPR)), specificity (True Negative Rate (TNR)) are calculated to assess the detailed performance. The true positive rate is higher, like 0.87, 0.95, and 0.95 for Access, Egress, and Interchange, which is well above the threshold.

#### 3.3.2. Encryption

The proposed encryption method is explained in Section 2.5 where encryption is applied on plain text images shown in Figure 4. Encryption results are highlighted in Figure 8. From encrypted results, one can see that plain text and encrypted information are different and an intruder cannot get information from the encrypted data. However, one visual inspection is not sufficient, and hence, we have evaluated the proposed scheme on several security parameters. Interested readers can find more details on these parameters in references [42,53,54]. Results are highlighted in Table 3, Table 4 and Table 5. Security of the proposed scheme is evident through lower correlation, homogeneity, and energy values. Furthermore, higher values of entropy, key sensitivity, number of pixel change rate (NPCR), unified average change intensity (UACI), and contrast also highlight higher security of the encrypted data in all cases of our AEI framework.

## 4. Conclusions

The novel spatio-temporal mobility prediction based optimisation and encryption algorithm proposed in this paper can solve numerous future 5G network pathways and traffic movement problems. The proposed AEI framework employs the ingenious concept of future passenger location estimations and accuracies through which advantages of futuristic optimisation can be maximised. It then devises classification of mobility prediction and preserving important passengers data with encryption for the estimated future network scenario. Most of the conventional approaches deal with reactive instead of predictive mode of operations expected to perform necessary optimisation in response to dynamic environment changes, which leaves behind a gap for the availability of computational resources. On the contrary, the proposed approach provides state-of-the-art heuristic techniques for traffic movement predictions, network pathway implementation, implementation of encrypted traffic flows, and practical solutions to address optimisation problems ahead of time in the underground train network. With this outset, 5G ambitions regarding address latency and QoS issues can be met. Therefore, it comprehensively discusses the proactive ML based classification of mobility prediction algorithms, mapping of optimal classification algorithms and an encryption algorithm with a single map for faster processing. Extensive simulations based on real-time traffic employing realistic classification of mobility predictions using AEI framework can achieve 91.17% with an ANN algorithm as compared to other ML algorithms mentioned in this paper. Comparative performance analysis with movement precision of train traffic flows indicate adequate stability and robustness of the proposed AEI framework towards predicting accuracies, and hence, it provides advanced encryption to the sensitive information to preserve from threats. Moreover, the AEI framework provides an opportunity to discuss cell load, coverage and capacity, and energy efficiency coupled with carbon emissions in the underground train network due to the overlap among their primary parameters of optimisation, which would be presented in our upcoming researches. For future works, we would provide metrics to implement optimisation with the incorporation of underground specific tap-in tap-out individual offsets and energy efficiency coupled with carbon emissions. Furthermore, we aim to work along the lines of user specific behaviours to maintain QoS requirements, provide reliable encryption and cellular cell constraints to effectively serve underground traffic flow. Another promising area of the research would incorporate several direction bounds of train traffic in conjunction to address energy efficiency and network pathway design problems.

## Figures and Tables

**Figure 1 sensors-20-02629-f001:**
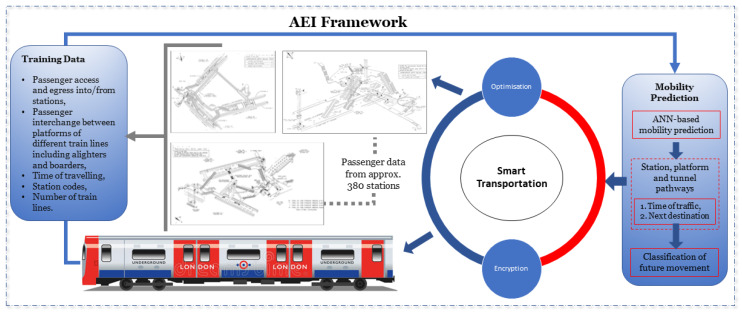
Access, Egress, and Interchange (AEI) Framework with data collection points from approximately 380 stations.

**Figure 2 sensors-20-02629-f002:**
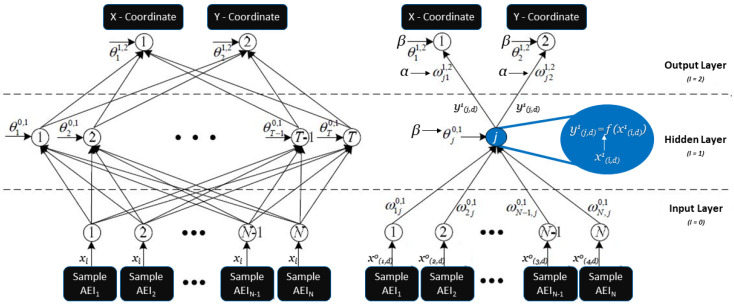
Topological mapping of Artificial Neural Network (ANN)-based mobility prediction for the London Underground and Overground (LUO) environment.

**Figure 3 sensors-20-02629-f003:**
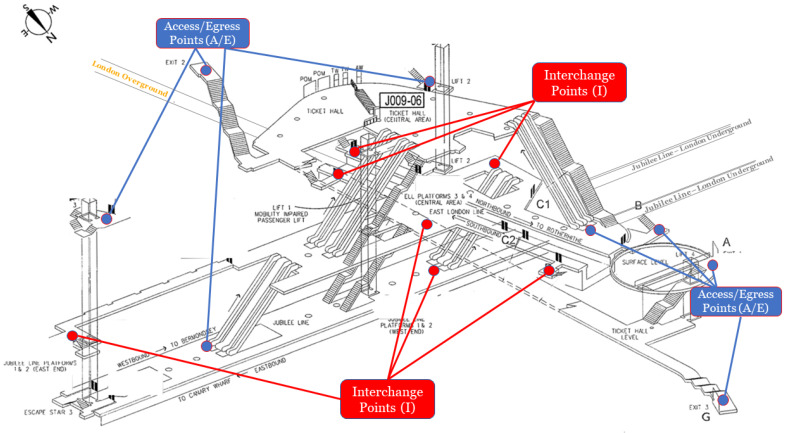
Test scenario layout.

**Figure 4 sensors-20-02629-f004:**
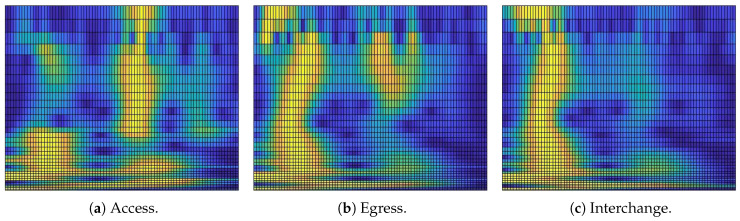
Plain text AEI data.

**Figure 5 sensors-20-02629-f005:**
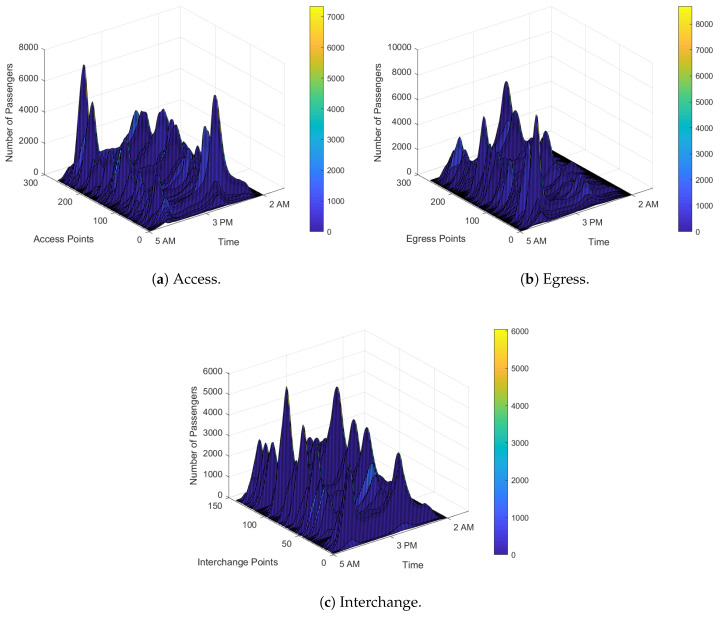
Simulation results for train traffic flows AEI framework considering three dimensional (3D) dataset for 1 week in the year 2017–2018. Plotted LUO dataset is from 05:00 AM to 02:00 AM (21-h) by different train lines on multiple stations with specific station numbers.

**Figure 6 sensors-20-02629-f006:**
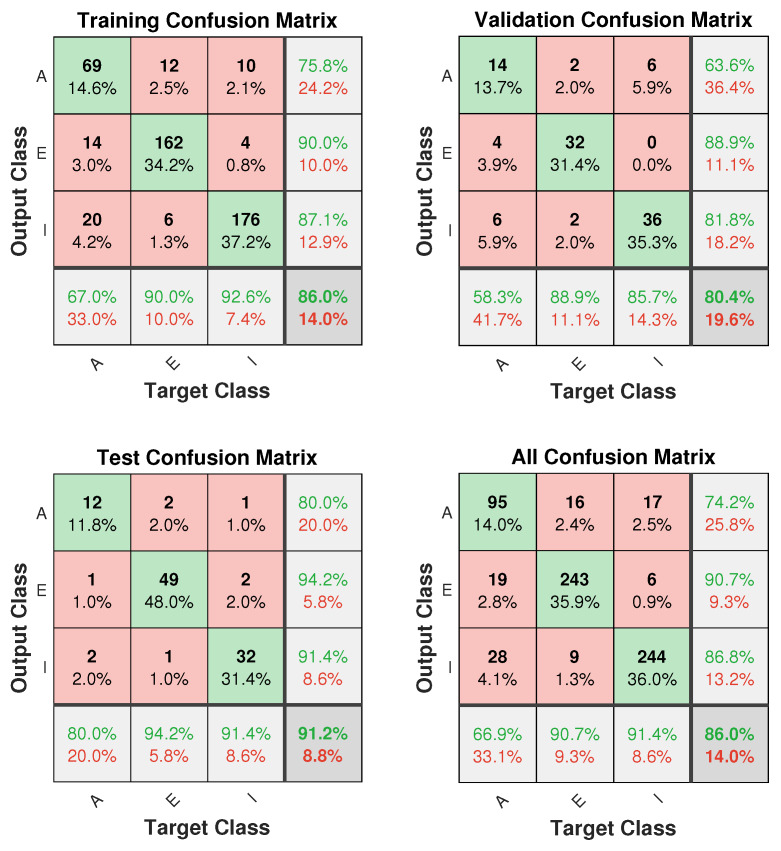
ANN algorithm based simulation results for the classification of train traffic flows AEI framework mobility predictions using three classes; Access (A), Egress (E), and Interchange (I).

**Figure 7 sensors-20-02629-f007:**
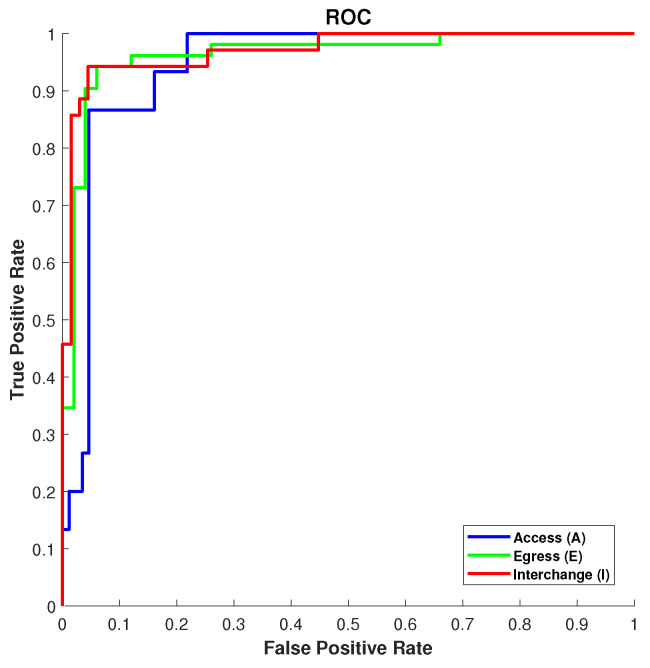
Receiver operating characteristic (ROC) curve for three classes Access (A), Egress (E), and Interchange (I) representing the specificity pair corresponding to a AEI Framework decision threshold. Grey line in the middle of the plot is a threshold line between True Positive and False Positive Ratios.

**Figure 8 sensors-20-02629-f008:**
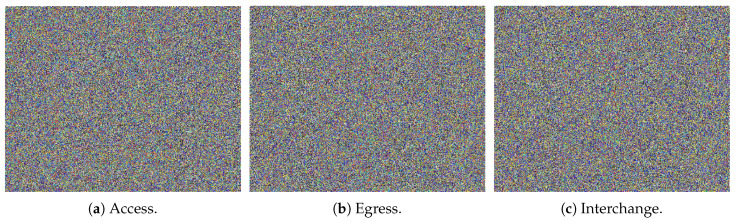
Encrypted text AEI data.

**Table 1 sensors-20-02629-t001:** Simulation scenario and settings (M = millions, h = hours).

Data Type	Value
Number of stations (Access/Egress)	380
Number of platforms (Interchanges)	270
Total number of passengers	12.15 M
Total number of passenger (Early)	0.41 M
Total number of passenger (AM Peak)	3.18 M
Total number of passenger (Midday)	3.02 M
Total number of passenger (PM Peak)	3.34 M
Total number of passenger (Evening)	1.51 M
Total number of passenger (Late)	0.68 M
Number of classes	3
Area of passenger movement probability	100%
Total simulation duration	21 h

**Table 2 sensors-20-02629-t002:** Classification of mobility prediction accuracies.

Machine Learning Algorithm	Accuracy
Artificial Neural Network (ANN)	91.17%
Discriminant Analysis (DT)	80.18%
K-Nearest Neighbour (KNN)	79.61%
Support Vector Machine (SVM)	79.47%
Decision Tree (DT)	76.37%
Naive Bayes (NB)	48.18%

**Table 3 sensors-20-02629-t003:** Access data.

Security Parameter	Plain Text	Encrypted Form
CorrCoff (H)	0.5890	−0.0021
CorrCoff (V)	0.7438	0.0032
CorrCoff (D)	0.4077	0.0193
Entropy	7.1273	7.7068
NPCR	NA	99.6314%
UACI	NA	33.4710
Contrast	3.7399	10.5089
Homogeneity	0.6809	0.3899
Energy	0.0854	0.0156

**Table 4 sensors-20-02629-t004:** Egress data.

Security Parameter	Plain Text	Encrypted Form
CorrCoff (H)	0.5579	0.0341
CorrCoff (V)	0.7593	0.0201
CorrCoff (D)	0.3981	0.0186
Entropy	7.1273	7.7068
NPCR	NA	99.6140%
UACI	NA	33.5032
Contrast	3.9899	10.4951
Homogeneity	0.6831	0.3881
Energy	0.0828	0.0156

**Table 5 sensors-20-02629-t005:** Egress data.

Security Parameter	Plain Text	Encrypted Form
CorrCoff (H)	0.6240	−0.0071
CorrCoff (V)	0.7705	−0.0309
CorrCoff (D)	0.4199	0.0190
Entropy	7.1273	7.7068
NPCR	NA	99.6338%
UACI	NA	33.4917
Contrast	3.5455	10.5004
Homogeneity	0.6922	0.3887
Energy	0.1042	0.0156

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
