# Peer review of "Mobility Prediction-Based Optimisation and Encryption of Passenger Traffic-Flows Using Machine Learning"

_sensors, 2020, doi:10.3390/s20092629_

Round 1
Reviewer 1 Report
A paragraph on the description of RFID sensor data collection may be considered.
Reviewer 2 Report
- The work discusses the mobility prediction based optimization which san solve the 5G network pathways.
- Similar research in the road mode done prior to that, however, this work highlights the rail mode, hence, it does not reduce the work novelty.
- 45 references have been give. With having such long work, at least 60 references would be suggested, which can balance the work volume, mainly in the introduction offered.
- System model described properly, specially in Fig 2 and Fig 3, and algorithms are easy to understand and follow, which is a merit for the work.
- ML in part 3 highlighted, but not in such a way that it should be, it is suggested to emphasize only the correlation between ML and mobility in a deeper way, also here is a good place to compensate the lack of reference. It can add to the value of the work.
- Result is given properly and defended accordingly.
- English language can be improved.
- In all I have no objection to its publication.
Reviewer 3 Report
The study is an interesting and timely one. The article contains information technical and innovative that justifies its publication. The problem addressed is current and has technical relevance, which makes it significant. The concept, methodology and results are well organized and clearly presented. The experimental methodology is described as comprehensively. Interpretations and conclusions are justified by the results.
My recommendations are:
- Quality of Figures is so important too. Please provide some high-resolution figures. The comparison of different methods using clear graphs should be explained;
- The author should depict the flow graph to illustrate the need of the proposed approach.
-
Literature review techniques has to be strengthened by including the issues in the current system and how the author proposes to overcome the same.
- Considering equations 1, 2 and 3 the meaning of index t and its possible variations;
-
Authors should add more details about the implementation of the code to perform the analysis and the library involved in this task.
